# ▲TRIFORCE: Lossless Acceleration of Long Sequence Generation with Hierarchical Speculative Decoding

**Hanshi Sun[1], Zhuoming Chen[1], Xinyu Yang[1], Yuandong Tian[2], Beidi Chen[1,2]**
[1]Carnegie Mellon Univeristy
[2]Meta AI (FAIR)
{hanshis, zhuominc, xinyuya2, beidic}@andrew.cmu.edu
yuandong@meta.com

## Abstract

With large language models (LLMs) widely deployed in long content generation recently, there has emerged an increasing demand for efficient long-sequence inference support. However, key-value (KV) cache, which is stored to avoid re-computation, has emerged as a critical bottleneck by growing linearly in size with the sequence length. Due to the auto-regressive nature of LLMs, the entire KV cache will be loaded for every generated token, resulting in low utilization of computational cores and high latency. While various compression methods for KV cache have been proposed to alleviate this issue, they suffer from degradation in generation quality. We introduce TRIFORCE, a hierarchical speculative decoding system that is scalable for long sequence generation. This approach leverages the original model weights and dynamic sparse KV cache via retrieval as a draft model, which serves as an intermediate layer in the hierarchy and is further speculated by a smaller model to reduce its drafting latency. TRI-FORCE not only facilitates impressive speedups for Llama2-7B-128K, achieving up to $2.31\times$ on an A100 GPU but also showcases scalability in handling even longer contexts. For the offloading setting on two RTX 4090 GPUs, TRIFORCE achieves 0.108s/token—only half as slow as the auto-regressive baseline on an A100, which attains $7.78\times$ on our optimized offloading system. Additionally, TRIFORCE performs $4.86\times$ than DeepSpeed-Zero-Inference on a single RTX 4090 GPU. TRIFORCE's robustness is highlighted by its consistently outstanding performance across various temperatures. The code is available at https://github.com/Infini-AI-Lab/TriForce.

## 1 Introduction

Large language models (LLMs) with long-context capability, such as GPT-4 (Achiam et al., 2023), Gemini (Team et al., 2023), and LWM (Liu et al., 2024a) continue to emerge and gain proficient application in scenarios including chatbots, vision generation, and financial analysis (Touvron et al., 2023; Chowdhery et al., 2023; Zhao et al., 2023; Reddy et al., 2024). However, losslessly serving these LLMs efficiently is challenging. Because of the auto-regressive nature of LLMs, the entire key-value (KV) cache, which stores intermediate key-value states from previous contexts to avoid re-computation, together with model parameters will be loaded into GPU SRAM for every token generated, resulting in low utilization of computational cores. In addition to the large volume of model parameters, the memory footprint of KV cache, which grows linearly with sequence length (Pope et al., 2023), is emerging as a new bottleneck for long sequence generation.

Recent methodologies have proposed KV cache eviction strategies (Xiao et al., 2023b; Zhang et al., 2024b; Liu et al., 2024c; Jiang et al., 2023; Ge et al., 2023) to mitigate the substantial memory footprint of KV cache, which selectively discard KV pairs from the cache based on a designed eviction policy, allowing models to generate texts with a limited KV cache budget. However, considering that discarded KV pairs cannot be restored and the difficulty in

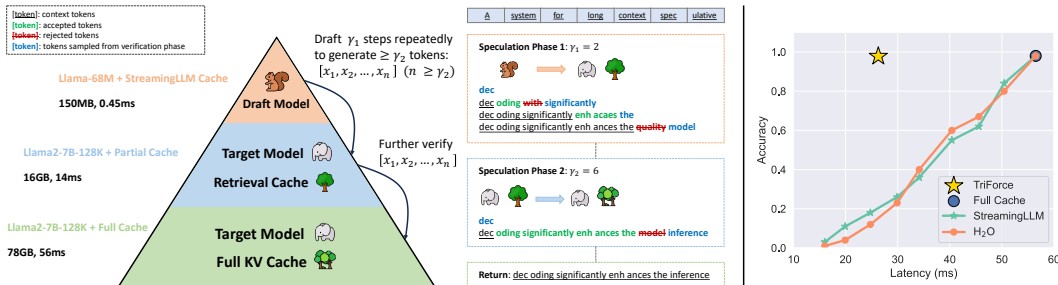

Figure 1: **Left**: TRIFORCE employs retrieval-based drafting and hierarchical speculation to effectively address the dual bottlenecks. It integrates two models and three caches, comprising a draft model, a target model, a StreamingLLM cache for the draft model, alongside a retrieval cache and a full cache for the target model. The process initiates by repeatedly drafting for $\gamma_1$ steps, assisting the target model with retrieved partial KV cache in generating over $\gamma_2$ tokens, which will be further verified by the target model using full KV cache. **Right**: Evaluating the Llama2-7B-128K on a needle retrieval task indicates that KV cache eviction-based methods, such as StreamingLLM, require a trade-off between latency and accuracy. In contrast, our TRIFORCE maintains low latency without sacrificing accuracy.

precisely foreseeing which KV pairs will be crucial for future text generation, they struggle with potential information loss, including hallucination and contextual incoherency (Yang et al., 2024), particularly in long contexts. Such challenges prevent these approaches from boosting speed without sacrificing the performance of models, as illustrated in Figure 1.

Concurrently, speculative decoding, which leverages a lightweight draft model to sequentially predict the next few tokens and let the target model verify the predicted tokens in parallel, is introduced to accelerate LLM inference while provably precisely preserving model output (Leviathan et al., 2023; Chen et al., 2023a; Xia et al., 2024). Nonetheless, deploying it for long sequence generation faces several challenges. First, training draft models to match the context length of target LLMs requires massive computation and it remains questionable whether these small models can achieve the same accuracy with a context length around 1M (Beltagy et al., 2020; Peng et al., 2023; Yan et al., 2024). Second, we found that draft models with existing training-free methods (e.g., KV cache eviction strategies) can result in poor speculating performance. A continuously increasing divergence (Leviathan et al., 2023) is witnessed as the sequence length increases, as shown in Figure 2a.

In pursuit of lossless acceleration, we utilize the lossless feature of speculative decoding as the foundation of our system. An ideal speculative decoding algorithm should (i) be training-free, (ii) maintain a high acceptance rate (Leviathan et al., 2023) with long contexts, and (iii) have low-cost drafting. However, two technical challenges need to be addressed to achieve the goal. First, it is not immediately apparent what we can use for low-latency drafting without training a smaller draft model to match the long context length. Second, the key factors for attaining a high acceptance rate with long contexts remain unclear.

Fortunately, based on our preliminary exploration, three key observations pave the way for designing an applicable system for serving LLMs with long contexts.

*Hierarchical Speculation for Dual Memory Bottlenecks*: As illustrated in Figures 2b and 2c, we recognize two memory bottlenecks: model weights and KV cache, and the latter gradually becomes the dominant bottleneck as context length increases. This inspires us to apply hierarchical speculation to tackle the two bottlenecks sequentially by different draft models.

*Leveraging Attention Sparsity for Speculative Decoding*: We identify considerable redundancy within KV cache, finding that a relatively small portion of it is sufficient to achieve a high acceptance rate by using partial KV cache as a draft cache for self-speculation.

*Exploiting Contextual Locality for Drafting Efficiency*: We discover that the information from long context tokens needed by adjacent tokens tends to be similar. This observation suggests

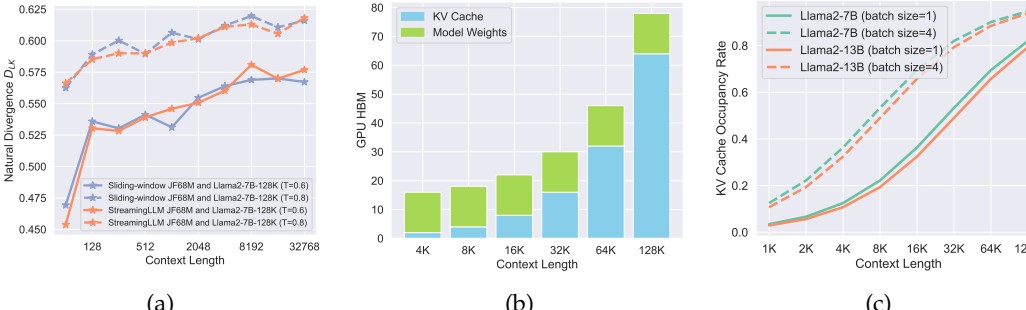

Figure 2: (a) A continuously increasing natural divergence (Leviathan et al., 2023) between the draft model with StreamingLLM or Sliding-window with Re-computation and Llama2-7B-128K is witnessed as the sequence length increases, indicating a falling acceptance rate for speculative decoding with longer contexts. Additionally, temperature sensitivity signals a lack of robustness. (b) Compared with model weights, KV cache gradually becomes another bottleneck with long contexts. (c) KV cache occupies most of the memory as the context length increases.

that a specific segment of the cache can be effectively reused across multiple decoding steps, amortizing the overhead of constructing draft cache and enhancing drafting efficiency.

Building on these insights, we introduce a hierarchical speculation approach. For a long-context target model (e.g., Llama2-7B-128K (Peng et al., 2023)), we leverage the original model weights but only with a small proportion (e.g., 3%) of KV cache as a draft to tackle the bottleneck of KV cache. Hierarchically, the draft model is further speculated by a lightweight model (e.g., Llama-68M) with StreamingLLM cache to address the bottleneck of model weights. We present TRIFORCE, depicted in Figure 1, a scalable and robust speculative decoding system that integrates retrieval-based drafting and hierarchical speculation, optimized for both on-chip and offloading scenarios. Specifically,

- In Section 4.1, by maintaining the full cache, we gain the flexibility to select KV pairs, allowing us to devise a superior selection method, termed retrieval-based drafting. This strategy retrieves required context information for future needs, which is characterized as lossless, particularly in comparison to eviction-based methods such as StreamingLLM and H$_2$O. We further demonstrated its effectiveness and robustness on different datasets.

- In Section 4.2, we propose a hierarchical system to address the dual memory bottlenecks. Using a lightweight model paired with a StreamingLLM cache for initial speculations, we can reduce the drafting latency for the subsequent speculation stage, thereby accelerating end-to-end inference.

Empirically, in Section 5, we perform extensive experiments and ablation studies to demonstrate the effectiveness of TRIFORCE. We show that TRIFORCE achieves up to 2.31× speedup for Llama2-7B-128K on a single A100 GPU on top of Hugging Face (Wolf et al., 2019) with CUDA graphs (NVIDIA & Fitzek, 2020). For the offloading setting, TRIFORCE attains an impressive 7.78× on two RTX 4090 GPUs, reaching 0.108 s/token—only half as slow as the auto-regressive baseline on an A100. TRIFORCE can efficiently serve a Llama2-13B with 128K contexts with 0.226s/token, which is 7.94× faster on our optimized offloading system. On a single RTX 4090 GPU, TRIFORCE is 4.86× faster than DeepSpeed-Zero-Inference (Aminabadi et al., 2022) [1]. Further, we show that: (i) TRIFORCE has a theoretical 13.1× upper bound, demonstrating exceptional scalability when dealing with long contexts; (ii) TRIFORCE is robust across various temperature settings, maintaining an acceptance rate above 0.9 even for a temperature of 1.0; and (iii) TRIFORCE's ability to efficiently process large batches, achieving a 1.9× speedup for a batch size of six with 19K contexts per sample.

---

[1]The official implementation of DeepSpeed-ZeRO-Inference (Aminabadi et al., 2022) with KV cache offloading currently only supports a single GPU, which computes attention on CPU. Our offloading system transfers KV cache from CPU to GPU, benefiting from Tensor Parallelism.

## 2 Background

### 2.1 Speculative Decoding

Speculative decoding (Stern et al., 2018; Leviathan et al., 2023; Chen et al., 2023a; Kim et al., 2024; Zhang et al., 2023; Santilli et al., 2023; Hooper et al., 2023) is featured by accelerating LLM decoding while precisely maintaining the model's output distribution. As the speed of the auto-regressive decoding process is mainly bound by the time for loading model weights and KV cache to GPU SRAM, speculative decoding leverages the observation that generating one token takes the same time as processing tens of tokens in parallel. Tree-based speculation methods are proposed to fully utilize the speculation budget (Fu et al., 2024; Li et al., 2024). Instead of making one prediction for the next token, tree-based methods leverage multiple candidates to boost the acceptance rate so that more tokens can get accepted (Miao et al., 2023; Sun et al., 2024; Chen et al., 2024). Staged speculation techniques (Spector & Re, 2023; Chen et al., 2023b) have been suggested to further accelerate inference by using a cascade of draft models, and hierarchical speculation shares similarities with these approaches. However, our method focuses on long sequence generation tasks, which presents unique challenges. We use different drafting methods for each speculation phase to address two distinct bottlenecks in long-context scenarios, instead of focusing on model weights at every stage for acceleration. Meanwhile, self-speculation approaches such as Medusa (Cai et al., 2024; Ankner et al., 2024), which are orthogonal to our method, require training efforts and can be integrated into our intermediate draft model.

### 2.2 KV Cache Eviction Strategies

**StreamingLLM** (Xiao et al., 2023b) addresses the limitations of window attention and sliding window with re-computation by presenting a straightforward yet effective method that allows LLMs to handle infinitely long text sequences without fine-tuning. StreamingLLM stabilizes the performance by retaining critical attention sink tokens together with recent KV for attention computation. By prioritizing sink tokens, StreamingLLM ensures the attention score distribution remains stable, promoting consistent language modeling for long texts.

**$H_2O$** (Zhang et al., 2024b) introduces a greedy but low-cost approach to processing infinite-length input streams, inspired by a simplified version of the heavy-hitters ($H_2$) eviction policy. This method dynamically updates the KV cache based on the cumulative attention scores, systematically removing the least critical KV to maintain a fixed cache size. By leveraging a greedy algorithm based on local statistics, $H_2O$ effectively selects which KV pairs to preserve in the cache, ensuring efficient inference without compromising quality.

However, it is important to recognize that these techniques do not increase the context window size (Zhang et al., 2024a; Jin et al., 2024; Ge et al., 2023; Jiang et al., 2023). They focus on retaining only the most recent tokens along with either attention sinks or heavy-hitters, while discarding other tokens. These approaches limit the model to processing based on their designed eviction policies and recent tokens. Consequently, they might not be directly applicable to tasks that demand comprehensive, long-context understanding.

### 2.3 KV Cache Quantization

Several approaches to KV cache quantization have been introduced to enhance the efficiency of inference for long sequence generation, aiming to maintain generation quality while reducing the memory consumption (Xiao et al., 2023a; Hooper et al., 2024; Sheng et al., 2023; Liu et al., 2023a; Zirui Liu et al., 2023; Yue et al., 2024). Quantization methods focus on compressing the bit width of KV cache activations, which is orthogonal to our approach.

## 3 Observation

Our design of TRIFORCE is inspired by two critical empirical observations regarding LLMs when dealing with long contexts, detailed as follows.

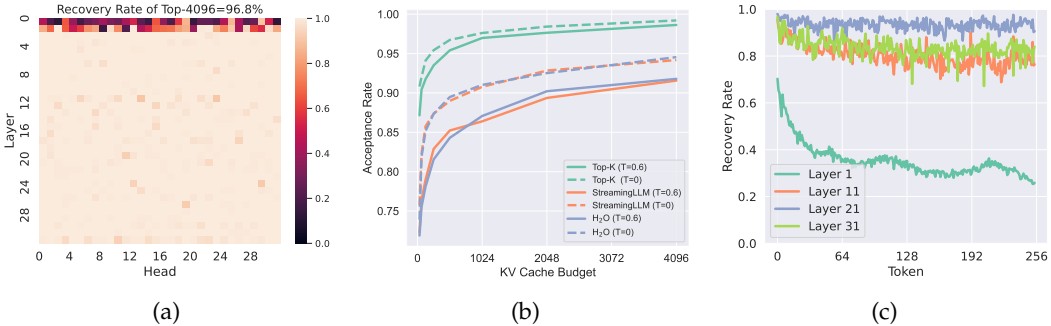

Figure 3: (a) The Llama2-7B-128K model demonstrates significant attention sparsity with a 120K context. Apart from the first two layers, the rest exhibit significant sparsity. (b) We can utilize partial KV cache and whole model weights to perform self-speculation. High acceptance rates are attainable using existing methods with a limited budget. (c) A notable degree of locality is observed in most layers, which gradually diminishes as context evolves.

## 3.1 Leveraging Attention Sparsity for Speculative Decoding

**Observation** The phenomenon of attention sparsity in pre-trained LLMs has been discovered by numerous studies (Zhang et al., 2024b; Xiao et al., 2023b; Liu et al., 2023b; 2024c). In our study, we conduct zero-shot inference on the PG-19 test set (Rae et al., 2019) with Llama2-7B-128K model. By visualizing the sparsity across different attention heads, demonstrated in Figure 3a, we observe that with a context length of 120K, it is possible to recover over 96% of the attention score with merely 4K tokens across almost all layers.

**Analysis** The presence of sparsity within the attention blocks suggests that a fraction of KV cache could serve as a draft cache to attain a high acceptance rate during self-speculative decoding. Since KV cache is the bottleneck under this setting, we can load whole model weights with partial KV cache as a draft model. Figure 3b demonstrates that utilizing only 1K tokens could theoretically achieve a 97.6% acceptance rate with Top-K selection method. While this scenario represents an optimal theoretical upper bound, practical implementations like $H_2O$ and StreamingLLM exhibit promising results, achieving over 90.5% acceptance rates with 1K KV cache budget. It should be noted that we maintain a full cache for the initial two layers for illustration purposes, while no layers are skipped in our practical system implementation for efficiency.

## 3.2 Exploiting Contextual Locality for Drafting Efficiency

**Observation** Our exploration reveals that the information from long context tokens needed by adjacent tokens tends to be similar. In our experiments, with the context length established at 120K, we instruct the model to generate 256 tokens. By choosing the top-4K indices according to the attention score of the last prefilled token, we use these indices to gather the attention scores for the subsequently generated tokens and assess the score's recovery rate for the initially prefilled 120K tokens. As shown in Figure 3c, it leads to high recovery across almost all layers and a slowly decreasing trend as the number of tokens increases.

**Insights** This observation allows for a single construction of the cache to suffice for multiple decoding steps, thereby amortizing the latency of constructing draft cache and boosting efficiency. As new KV cache are introduced, guided by the understanding that recent words are more strongly correlated with the tokens currently being decoded, these entries will replace the less significant ones. Cache re-building operations can be scheduled at regular intervals or adaptively in response to a drop in the acceptance rate, which ensures that the cache remains dynamically aligned with the evolving context. Notably, both StreamingLLM and $H_2O$ incorporate this principle implicitly. $H_2O$ consistently retains tokens with high scores, and StreamingLLM reuses extensive local information and sink tokens, which both reduce the necessity for complete cache reconstruction.

## 4 TRIFORCE

This section aims to introduce the TRIFORCE, which leverages a retrieval-based KV cache selection policy and a hierarchical speculation system. We first argue that our retrieval-based drafting approach is intuitive and lossless compared to existing strategies such as StreamingLLM and $H_2O$. Subsequently, we introduce the hierarchical system designed to effectively address the dual bottlenecks in speculative decoding, facilitating a substantial improvement in overall speed-up. Finally, TRIFORCE is elaborated in Section 4.3.

### 4.1 Retrieval-based Drafting

In scenarios requiring long-term contextual dependencies, methods like StreamingLLM and $H_2O$ underperform due to their cache updating strategies, which are ineffective at accurately retrieving detailed contextual information because they inevitably and irrecoverably discard KV pairs. In our experiment, we challenge StreamingLLM and $H_2O$ with a needle retrieval task (Liu et al., 2024b; Peng et al., 2023; Liu et al., 2024a). As detailed in Table 1, there is a notable drop in their acceptance rates compared to their performance on the PG-19 dataset, highlighting their limitations. Essentially, StreamingLLM and $H_2O$ operate on a lossy principle, as evicted tokens are permanently discarded, making them a poor fit for settings requiring the preservation of full KV cache for the target model.

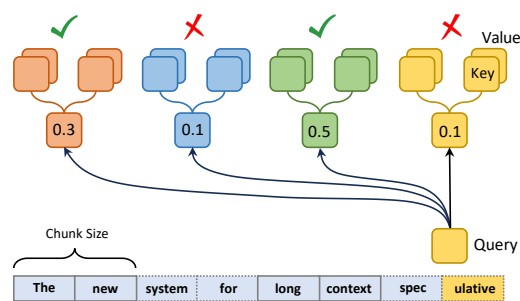

Figure 4: Retrieval-based drafting

The necessity of keeping the entire KV cache in our settings allows us to select KV cache more freely (Singhal et al., 2001). This insight leads us to develop a more effective selection policy for lossless approximations. In our approach, demonstrated in Figure 4, KV cache is segmented into small chunks. During the retrieval phase, we calculate the attention between a given query and the average key cache within each chunk. This method effectively highlights the most relevant chunks, enabling us to gather KV cache with a fixed budget based on the scores. As illustrated in Table 1, retrieval-based method excels by actively identifying the most crucial information for the task rather than relying on passive and time-based cache management methods. By focusing on relevance over recency, retrieval-based policy demonstrates its potential to handle contextually dense datasets.

Table 1: Acceptance rates are shown across various tasks, utilizing a 120K context and a 4K budget, while bypassing the initial two layers. There is a notable drop in StreamingLLM and $H_2O$'s results on needle retrieval. For reference, Top-K is the theoretical upper bound.

| Method | Top-K (Ref.) | StreamingLLM | $H_2O$ | Retrieval |
|---|---|---|---|---|
| PG-19 | 0.9921 | 0.9156 | 0.9179 | **0.9649** |
| Needle Retrieval | 0.9989 | 0.0519 | 0.0739 | **0.9878** |

### 4.2 Hierarchical Speculation

While addressing the KV cache bottleneck enhances efficiency, the requirement to load whole model weights for drafting reintroduces latency, shifting the bottleneck to model weights again. To tackle this challenge, we implement a hierarchical system, as illustrated in Figure 1. This system employs a secondary, lightweight model with StreamingLLM cache to perform initial speculations for the target model with retrieval-based cache (which serves as a draft model for the target model with full KV cache). By establishing this sequential speculation hierarchy, we effectively reduce drafting latency and accelerate overall inference.

**Correctness**: The original output distribution is preserved during the final speculation phase, which is identical to the standard speculative decoding algorithm (Leviathan et al., 2023; Chen et al., 2023a), and the proof is trivial.

### 4.3 Algorithm

TRIFORCE is devised to exploit the bottlenecks associated with both model weights and KV cache to enhance the inference speed of LLMs for long sequence generation. We present the pseudocode for the TRIFORCE in Algorithm 1. It starts by prefilling the target model $M_p$ with full cache $C_p$ and draft model $M_q$ with StreamingLLM cache $C_q$ using a given input prefix, and then constructs the retrieval cache $C_r$. The initialization and update mechanism for the retrieval cache $C_r$ is guided by the insights of contextual locality discussed in Section 3.2. We first construct $C_r$ using the last token of the prefix, arranging tokens by the descending order of importance. In subsequent inferences, we overwrite tokens with the least importance, maintaining the relevance and utility of the cache. A reconstruction of $C_r$ is triggered either when the rolling average acceptance rate drops below a threshold or at a designed stride.

The inference progresses iteratively until it reaches the target sequence length $T$. After each iteration, cache $C_r$ and $C_q$ are updated to prepare for the subsequent speculation phase. Each iteration encompasses two speculations: initially, $M_q$ utilizes $C_q$ to predict $M_p$ with $C_r$ for $\gamma_1$ steps until $n \geq \gamma_2$. Subsequently, these $n$ tokens are self-verified (Zhang et al., 2023) by $M_p$ with $C_p$. This process constructs a hierarchy: the first layer of hierarchy employs a smaller, faster model $M_q$ with local context $C_q$ to speculate the large model $M_p$ with partial but high-quality global context $C_r$, addressing the model weights bottleneck. The second layer utilizes model $M_p$ with retrieval cache for self-speculation, overcoming the bottleneck caused by KV cache. This hierarchical speculation algorithm boosts efficiency by effectively addressing both bottlenecks. System implementation is detailed in Appendix A.

---

**Algorithm 1 ♠▲ TRIFORCE**

---

1: **Input:** Prefix $[x_1, \cdots, x_t]$, target model $M_p$ with full cache $C_p$, draft model $M_q$ with StreamingLLM cache $C_q$, target sequence length $T$, speculation length $\gamma_1, \gamma_2$, drafting phase DRAFT, verification phase VERIFY, and correction phase CORRECT;
2: **Initialize:** Prefill $M_p$, $M_q$, construct retrieval cache $C_r$ using $x_t$, $N \leftarrow t$
3: **while** $N < T$ **do**
4:     $n \leftarrow 0$
5:     **while** $n < \gamma_2$ **do**
6:         Set $q_1, \cdots, q_{\gamma_1} \leftarrow \text{DRAFT}(M_q, C_q, x_{\leq N})$        ▷ Run $M_q$ with eviction cache $C_q$
7:         Sample $\tilde{x}_i \sim q_i, i = 1, \cdots, \gamma_1$
8:         Set $\hat{p}_1, \cdots, \hat{p}_{\gamma_1+1} \leftarrow M_p(C_r, x_{\leq N}, \tilde{x}_{\leq \gamma_1})$        ▷ Run $M_p$ with retrieval cache $C_r$
9:         **for** $i = 1$ **to** $\gamma_1$ **do**
10:             **if** $\text{VERIFY}(\tilde{x}_i, q_i, \hat{p}_i)$ **then**
11:                 $\hat{x}_{n+i} \leftarrow \tilde{x}_i$ and $n \leftarrow n + 1$
12:             **else**
13:                 $\hat{x}_{n+i} \leftarrow \text{CORRECT}(q_i, \hat{p}_i)$ and $n \leftarrow n + 1$
14:                 Break
15:             **end if**
16:         **end for**
17:         If all drafted tokens are accepted, sample next token $\hat{x}_{n+1} \sim \hat{p}_{\gamma_1+1}$ and $n \leftarrow n + 1$
18:     **end while**
19:     Collect $\hat{p}_1, \cdots, \hat{p}_n$ for $\hat{x}_1, \cdots, \hat{x}_n$
20:     Set $p_1, \cdots, p_{n+1} \leftarrow M_p(C_p, x_{\leq N}, \hat{x}_{\leq n})$        ▷ Run $M_p$ with full cache $C_p$
21:     **for** $i = 1$ **to** $n$ **do**
22:         **if** $\text{VERIFY}(\hat{x}_i, \hat{p}_i, p_i)$ **then**
23:             $x_{N+i} \leftarrow \hat{x}_i$ and $N \leftarrow N + 1$
24:         **else**
25:             $x_{N+i} \leftarrow \text{CORRECT}(\hat{p}_i, p_i)$ and $N \leftarrow N + 1$
26:             Break
27:         **end if**
28:     **end for**
29:     If all drafted tokens are accepted, sample next token $x_{N+1} \sim p_{n+1}$ and $N \leftarrow N + 1$
30:     Update $C_r$, $C_q$ based on the accepted tokens ▷ Update KV cache for the next iteration
31: **end while**

---

Table 2: **On-chip results (A100)**: We indicate the average acceptance rate in parentheses alongside the speedup factor. T means sampling temperature. In the A100 on-chip experiments, with a prompt length of 122K, and a generation length of 256, we evaluate TRIFORCE against the JF68M model with StreamingLLM cache (Naive Policy). The results clearly demonstrate that TRIFORCE significantly surpasses its performance.

| Method | T | Speedup | Naive Policy |
|---|---|---|---|
| TRIFORCE | 0.0 | **2.31**× (0.9234) | 1.56× (0.4649) |
| TRIFORCE | 0.2 | **2.25**× (0.9203) | 1.54× (0.4452) |
| TRIFORCE | 0.4 | **2.20**× (0.9142) | 1.47× (0.4256) |
| TRIFORCE | 0.6 | **2.19**× (0.9137) | 1.42× (0.4036) |
| TRIFORCE | 0.8 | **2.08**× (0.8986) | 1.34× (0.3131) |
| TRIFORCE | 1.0 | **2.08**× (0.9004) | 1.29× (0.2872) |
| TRIFORCE | 1.2 | **2.02**× (0.8902) | 1.27× (0.2664) |
| Retrieval w/o Hierarchy | 0.6 | 1.80× (0.9126) | - |
| StreamingLLM w/ Hierarchy | 0.6 | 1.90× (0.8745) | - |

Table 3: **Offloading results (RTX 4090)**: We present latency comparison between TRIFORCE and Auto-regressive (AR) baseline for various models on different GPU setups. The sampling temperature is set to 0.6. The results indicate that TRIFORCE achieves significant speedups across a range of models and hardware configurations. The entries marked with an asterisk represent the baseline using DeepSpeed-ZeRO-Inference (Aminabadi et al., 2022).

| GPUs | Target Model | TRIFORCE (ms) | AR (ms) | Speedup |
|---|---|---|---|---|
| 2× RTX 4090s | Llama2-7B-128K | 108 | 840 | 7.78× |
| 2× RTX 4090s | LWM-Text-Chat-128K | 114 | 840 | 7.37× |
| 2× RTX 4090s | Llama2-13B-128K | 226 | 1794 | 7.94× |
| 1× RTX 4090 | Llama2-7B-128K | 312 | 1516* | 4.86× |
| 1× RTX 4090 | LWM-Text-Chat-128K | 314 | 1516* | 4.83× |

## 5 Empirical Evaluation

In this section, our goal is to showcase the capabilities of TRIFORCE, a scalable and robust speculative decoding algorithm designed to expedite the inference of LLMs for long sequence generation, which significantly reduces the wall-clock time. We first present our end-to-end system, highlighting the overall speedup achieved, including both on-chip and offloading settings, followed by a comparison with other methods and ablation experiments.

### 5.1 End-to-end Results

We demonstrate that TRIFORCE accelerates long sequence generation, up to 2.31× on an A100 in the on-chip setting and 7.78× on two RTX 4090s with offloading for Llama2-7B-128K.

**Setup.** Our experiments are based on Llama2 and LWM models with 128K context window size (Touvron et al., 2023; Liu et al., 2024a; Peng et al., 2023), which serve as our target models. In this setup, we utilize a 4K retrieval cache as an intermediate draft cache in our hierarchical system, while leveraging the JackFram/Llama68M (JF68M) (Miao et al., 2023) model as the initial draft model. For experiments involving offloading, we aim to maximize memory utilization by filling it up as much as possible and offloading the remaining KV cache to the CPU (AMD EPYC 9754 @ 2.25 GHz), while keeping the model weights on the GPU. Our evaluation is carried out on the PG-19 (Rae et al., 2019) and NarrativeQA (Kočiský et al., 2018) dataset, each testing on 100 examples, configured to a prompt length of 122K for on-chip settings and 127K for offloading settings, and aiming for a generation of 256 tokens. The performance of TRIFORCE is analyzed across various hardware configurations, including on-chip experiments on an A100, and offloading experiments on RTX 4090 GPUs.

**Naive Policy.** Since it is hard to train a draft model with long contexts, we consider JF68M with StreamingLLM cache as a naive policy approach, and its budget is set to 1K. Additionally, we experiment with various temperatures to test its robustness.

**Main Results.** We evaluate TRIFORCE using different temperatures, as depicted in Table 2. We observe that TRI-FORCE reaches up to 2.31× speedup for the on-chip setting with a minimal 4K KV cache budget for Llama2-7B-128K. For offloading settings, we provide end-to-end results on consumer GPUs for more models, including Llama2-7B-128K, Llama2-13B-128K, and LWM-Text-Chat-128K. Remarkably, in Table 3 we demonstrate that TRIFORCE can efficiently serve a Llama2-13B with 128K contexts on two RTX 4090s, reaching an average time between tokens as low as 0.226 seconds, which is 7.94× faster than a highly optimized offloading system. Moreover, with TRIFORCE, Llama2-7B-128K can be served with 0.108s/token—only half as slow as the auto-regressive baseline on an A100. We

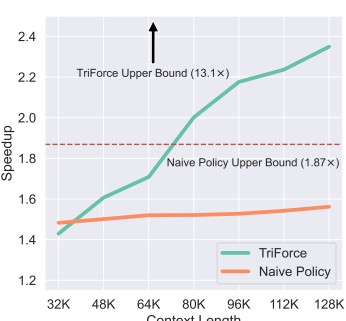

Figure 5: TRIFORCE's excellent scalability with longer contexts

also illustrate how TRIFORCE boosts the efficiency of batched inference, a more frequently employed setting in real-world model serving. TRIFORCE achieves 1.9× for a batch size of six, with each sample in the batch having 19K contexts, which is demonstrated in Table 4.

Table 4: **Batching results (A100)**: TRIFORCE showcases its exceptional capability in efficiently handling large batch sizes, consistently exceeding the performance of the JF68M model with StreamingLLM cache across all configurations for Llama2-7B-128K.

| Batch | Budget | T | Speedup | Naive Policy |
|---|---|---|---|---|
| (2,56K) | (2,1024) | 0.0 | **1.89×** | 1.46× |
| (2,56K) | (2,1024) | 0.6 | **1.75×** | 1.35× |
| (6,19K) | (6,768) | 0.0 | **1.90×** | 1.39× |
| (6,19K) | (6,768) | 0.6 | **1.76×** | 1.28× |
| (10,12K) | (10,768) | 0.0 | **1.72×** | 1.34× |
| (10,12K) | (10,768) | 0.6 | **1.61×** | 1.21× |

**Analysis.** (1) Effectiveness: TRIFORCE's integration of the hierarchical system significantly enhances speedup, with TRIFORCE showing marked improvements over both the StreamingLLM method with hierarchical speculation and retrieval method without the hierarchical system. (2) Scalability: As depicted in Figure 5, TRIFORCE demonstrates excellent scalability with longer context lengths. This scalability is attributed to its high acceptance rate and the growing gap between the draft and the target model's latencies. Theoretically, TRIFORCE could achieve a speedup of up to 13.1×, 7 times higher than the naive policy, underscoring its significant scaling potential. (3) Robustness: Unlike vanilla speculative decoding methods, TRIFORCE maintains relatively consistent performance across various temperature settings. It exhibits less temperature sensitivity, maintaining an acceptance rate above 0.9 even when the temperature is set to 1.0, highlighting its stability and reliability.

### 5.2 Comparison with Other Methods

We provide a comparison with REST (He et al., 2023) and Skipping Layers (Zhang et al., 2023). Table 5 compares TRIFORCE, REST, and Skipping Layers with Llama2-7B-128K on an A100 using PG-19 dataset, showing TRIFORCE achieves the best speedup for long sequence generation.

| Method | Speedup |
|---|---|
| TRIFORCE | **2.31×** |
| REST | 1.47× |
| Skipping Layers | 1.36× |

Table 5: Speedup comparison with REST and Skipping Layers

TRIFORCE retrieves information from the KV cache, allowing dynamic adaptation to contexts, while REST uses an external predefined datastore. In Table 5, Skipping Layers utilizes 68% of the KV cache, whereas TRIFORCE efficiently uses only 3%, addressing the bottleneck in long-context scenarios better.

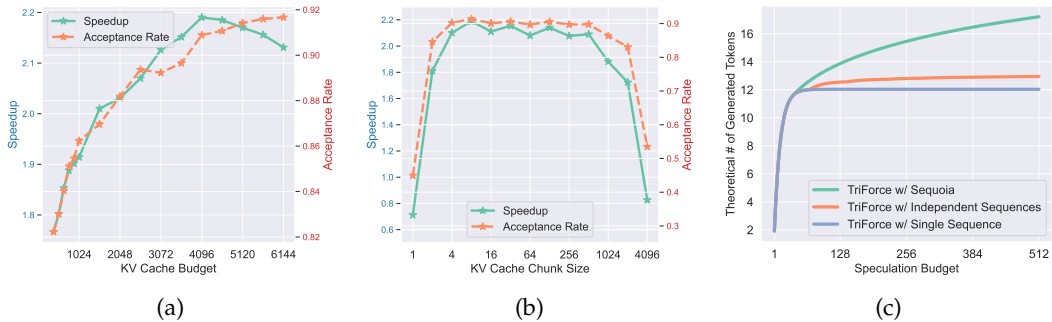

Figure 6: (a) Analyzing speedup and acceptance rates across different KV cache budgets reveals that a 4K budget is optimal, balancing acceptance rates and the drafting overhead. (b) For a 4K KV cache budget, excessively small chunk sizes may overfit individual tokens, while overly large chunk sizes could limit selection diversity. (c) TRIFORCE is compatible with tree-based speculations, enhancing the theoretical average number of tokens generated per decoding step of the target model by employing larger speculation budgets.

### 5.3 Ablation Results

We present extensive ablation studies of TRIFORCE, focusing on three key points: (1) the influence of different KV cache budgets, (2) the impact of chunk size selection, and (3) TRIFORCE's compatibility with tree-based speculative decoding.

#### 5.3.1 KV Cache Budget

As illustrated in Figure 6a, for Llama2-7B-128K, the acceptance rate rises with the cache budget up to 4K, then plateaus towards 1.0. This suggests that increasing the cache size beyond 4K offers diminishing benefits due to drafting latency. Thus, a 4K KV cache budget is optimal for TRIFORCE, balancing high acceptance rates and minimal drafting overhead.

#### 5.3.2 KV Cache Chunk Size

Since we utilize contextual locality to reuse the retrieval cache, we need to examine the impact of KV cache chunk size on performance. Figure 6b shows that smaller chunks may overfit to single tokens, limiting generalization, while larger chunks may dilute high-score tokens with low-score ones, resulting in reduced differentiation among chunks. Large chunks also reduce selection flexibility, constraining diversity within a fixed cache budget.

#### 5.3.3 Compatibility with Tree-based Speculative Decoding

We explore the possibility of integrating TRIFORCE with tree-based speculative decoding. Specifically, for Llama2-7B-128K on an A100, we estimate the theoretical number of generated tokens when TRIFORCE is combined with tree structures, including Sequoia (Chen et al., 2024) and Independent Sequences. As depicted in Figure 6c, this integration can potentially improve the end-to-end speedup by utilizing additional speculation budgets.

## 6 Conclusion

In this work, we introduced TRIFORCE, a hierarchical speculative decoding system aimed at significantly enhancing the efficiency of serving LLMs with long contexts. Leveraging insights from attention sparsity and contextual locality, TRIFORCE mitigates the dual bottlenecks associated with KV cache and model weights. Our empirical experiments demonstrate TRIFORCE's remarkable performance, including a notable speedup of up to 2.31× on an A100 and 7.78× on two RTX 4090s with offloading, achieving 0.108s/token—only half as slow as the auto-regressive baseline on an A100. These achievements illustrate TRIFORCE's potential to revolutionize the serving of long-context models for long sequence generation.

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

**Acknowledgments**

We thank Yang Zhou, Ranajoy Sadhukhan, Harry Dong, and Jian Chen for their helpful discussions and feedback on early drafts of the paper.

# A  System Implementation

We implement the draft and target models using Transformers (Wolf et al., 2019). Leveraging a predetermined cache budget enables the effective use of PyTorch CUDA graphs (Paszke et al., 2019; NVIDIA & Fitzek, 2020), significantly minimizing the kernel launching overhead during speculative decoding. FlashAttention is used to accelerate attention operations (Dao et al., 2022; Dao, 2023; Kwon et al., 2023; Hong et al., 2023). Notably, we maintain full layer sparsity without omitting the initial two layers for system efficiency, ensuring our approach stays within the fixed KV cache budget. Although this strategy might lead to a lower acceptance rate, the benefit of maintaining a constant drafting cost makes it a worthwhile compromise. Additionally, to facilitate a faster speculation phase, we also implement two extra speculation cache structures, including our retrieval cache and StreamingLLM cache.

# B  Additional Experiments

## B.1  TRIFORCE's Scalability for Longer Inputs

For longer inputs, a solution is offloading the KV cache to the CPU memory. Below are experiments on a single L40 GPU with offloading for longer inputs on LWM-Text models (Liu et al., 2024a) using the PG-19 dataset, showing impressive speedup.

| Model | Speedup |
|---|---|
| LWM-Text-256K | 11.81× |
| LWM-Text-512K | 12.10× |

Table 6: TRIFORCE's Scalability for longer inputs

## B.2  TRIFORCE's Scalability for Longer Outputs

In the previous sections of the paper, we focused on a long input and a short output to illustrate the efficiency gains in a commonly encountered setting. Here we provide additional experiments with longer output sequences up to 2K with Llama2-7B-128K on an A100 GPU using the PG-19 dataset:

| Output Sequence Length | Speedup |
|---|---|
| 256 | 2.31× |
| 512 | 2.28× |
| 768 | 2.34× |
| 1024 | 2.32× |
| 1536 | 2.28× |
| 2048 | 2.29× |

Table 7: TRIFORCE's Scalability for longer outputs

As seen in Table 7, the performance of TRIFORCE remains relatively stable. This stability is due to the retrieval cache being reconstructed at fixed intervals, ensuring it stays aligned with the evolving context. Furthermore, our retrieved cache can be effectively reused across multiple decoding steps, amortizing the overhead of retrieval. These results demonstrate the scalability of TRIFORCE and its robustness in handling varying output lengths, thereby addressing the concern about its generalizability.

## B.3  Ablation of $\gamma_1, \gamma_2$

Here we provide additional ablation results of $\gamma_1, \gamma_2$ in Table 8 with Llama2-7B-128K on an A100 GPU using the PG-19 dataset. Due to the gap between JF68M with StreamingLLM

and Llama-7B-128K with retrieval cache, the acceptance rate of the first speculation phase in our speculation hierarchy is low. As shown in the table, the optimal value for $\gamma_1$ is 2 and for $\gamma_2$ is 6.

| $\gamma_1$ | Speedup |
|---|---|
| 1 | 2.30× |
| **2** | **2.31×** |
| 3 | 2.24× |
| 4 | 2.17× |
| 5 | 2.09× |
| 6 | 2.01× |
| 7 | 1.92× |
| 8 | 1.83× |

| $\gamma_2$ | Speedup |
|---|---|
| 1 | 1.62× |
| 2 | 1.90× |
| 3 | 2.13× |
| 4 | 2.22× |
| 5 | 2.26× |
| **6** | **2.31×** |
| 7 | 2.24× |
| 8 | 2.20× |

(a) Speedup at different $\gamma_1$ values with $\gamma_2 = 6$     (b) Speedup at different $\gamma_2$ values with $\gamma_2 = 2$

Table 8: Ablation of $\gamma_1, \gamma_2$

