# OpenReview forum: "TriForce: Lossless Acceleration of Long Sequence Generation with Hierarchical Speculative Decoding"
_colmweb.org/COLM/2024/Conference — COLM_

### Official Review · Reviewer_FT2L · 2024-05-11

**Rating:** 7
**Confidence:** 4
**Ethics Flag:** 1

**Summary:**

This research introduces TRIFORCE, a novel hierarchical speculative decoding system that significantly accelerates inference efficiency for large language models on long context settings while fully preserving output quality. By leveraging insights from attention sparsity and contextual locality, the authors propose innovative techniques such as retrieval-based drafting (large model with retrieved cache) and hierarchical speculation (small model with StreamingLLM cache) to effectively mitigate the dual memory bottlenecks posed by KV cache and model weights. Comprehensive experiments on the PG-19 dataset demonstrate the effectiveness and practicality of the proposed approach, achieving up to 2.31x speedup on A100 GPUs and 11.65x speedup in offloading scenarios, as well as strong performance on large-batch inference tasks.

**Reasons To Accept:**

1. Insightful observations in Sections 3.1 and 3.2 effectively motivate the design of TRIFORCE's key techniques.

2. TRIFORCE consistently outperforms existing methods in long-context settings, demonstrating its effectiveness in efficiently handling long sequences.

**Reasons To Reject:**

1. The evaluation focuses on a single setting with a long input (120K tokens) and a short output (256 tokens). To better understand TRIFORCE's generalizability, it would be valuable to test its performance on scenarios with longer output sequences and analyze how the method scales with increasing output length.

2. The paper does not specify the hyperparameters γ1 and γ2 values, which control the speculation lengths in the hierarchical system. To facilitate reproducibility and deeper understanding, it is important to list all the hyperparameters used in the experiments and discuss their impact on the system's performance, as the effects of these hyperparameters remain unclear.

---

> ### Author Rebuttal · Authors · 2024-05-31
>
> Thank you for the supportive comments and recognizing the novelty of our method and the thorough evaluations. We hope our detailed clarifications and additional experimental results will address the concerns regarding our work.
>
> **Q1: Performance with longer outputs and scalability with increasing length**
>
>
> We appreciate the suggestion to evaluate TriForce on scenarios with longer output sequences. In our current experiments, we focused on a long input and a short output to illustrate the efficiency gains in a commonly encountered setting. Here we provide additional experiments with longer output sequences up to 2K with Llama2-7B-128K on an A100 GPU using the PG-19 dataset:
>
> | Output Sequence Length | Speedup |
> | - | - |
> |256|2.31x|
> |512|2.28x|
> |768|2.34x|
> |1024|2.32x|
> |1536|2.28x|
> |2048|2.29x|
>
> As seen in the table, performance remains relatively stable. This stability is due to the retrieval cache being reconstructed at fixed intervals, ensuring it stays aligned with the evolving context. Furthermore, our retrieved cache can be effectively reused across multiple decoding steps, amortizing the overhead of retrieval. These results demonstrate the scalability of TriForce and its robustness in handling varying output lengths, thereby addressing the concern about its generalizability.
>
> **Q2: Ablation of $\gamma_1, \gamma_2$**
>
>
> Thank you for the suggestion. Here are the values used in our experiments: $\gamma_1=2, \gamma_2=6$. We also provide additional ablation results below with Llama2-7B-128K on an A100 GPU using the PG-19 dataset. We will also release our code to the public to facilitate the reproducibility of our work.
>
>
> Ablation of $\gamma_1$ ($\gamma_2=6$): Due to the gap between JF68M with StreamingLLM and Llama-7B-128K with retrieval cache, the acceptance rate of the first speculation phase in our speculation hierarchy is low. As shown in the table, the optimal value for $\gamma_1$ is 2.
> |$\gamma_1$|Speedup|
> | - | - |
> |1|2.30x|
> |**2**|**2.31x**|
> |3|2.24x|
> |4|2.17x|
> |5|2.09x|
> |6|2.01x|
> |7|1.92x|
> |8|1.83x|
>
> Ablation of $\gamma_2$ ($\gamma_1=2$): The table below shows that $\gamma_2=6$ is the optimal value observed.
> |$\gamma_2$|Speedup|
> | - | - |
> |1|1.62x|
> |2|1.90x|
> |3|2.13x|
> |4|2.22x|
> |5|2.26x|
> |**6**|**2.31x**|
> |7|2.24x|
> |8|2.20x|
> |9|2.17x|
> |10|2.13x|
>
> We hope these additions address your concerns about the generalizability and reproducibility of our work. Let me know if there are any specific details or further adjustments you’d like to make!

---

### Official Review · Reviewer_myEE · 2024-05-16

**Rating:** 6
**Confidence:** 4
**Ethics Flag:** 1

**Summary:**

The paper focuses on improving the decoding efficiency of language models on long context generation, where KV cache IO can be a major bottleneck. The goal is to improve the generation latency without sacrificing accuracy of the language models. The paper proposes TriForce, a speculative decoding method. TriForce is a hierarchical speculative decoding method, where there are two draft models, one is based on a small language model with StreamingLLM KV cache, and the other is based on the original target model but with a small subset of KV cache (obtained by retrieval). It can achieve more than 2x speedup when context is long (e.g., >120K tokens).

**Questions To Authors:**

- In Figure 3(b): what does “Top-K” mean? Is this to take the subset of KV cache with top-k largest attention weights? Also, for “acceptance rate”: I would suggest defining what it is before talking about the results.
- What do you mean by “maintain a full cache for the initial two layers”?
- In Figure 3(c): what is the x-axis? What does “recovery rate” mean?
- Table 2/3: What is “JF68M Eviction”?

**Reasons To Accept:**

- Lossless efficient decoding for long-context generation, I believe, has become more and more important. It is good to study this problem, as it is very different scenario compared to what people used to study widely (e.g., speculative decoding with less than one thousand tokens).
- The general idea that using the model with a subset of KV cache as a draft model makes sense to me, especially in the case of long-context generation.
- The method is well-motivated by a couple of great observations (section 3 and figure 3). Those observations can be insightful themself, and facilitate other research.
- The idea of selecting KV cache via retrieval is valid. It is also good that the paper shows that existing methods (StreamingLLM and H2O) perform poorly on the needle retrieval task.
- The experimental results are fairly strong.

**Reasons To Reject:**

- The presentation of the paper can be improved. There are quite a few places, which can be hard to follow without being very familiar with the related work/concepts. (I list some of them in the “Questions” section)
- This approach requires a small draft model for the hierarchical speculation. This may not easily exist and may also take other resources during inference.
- I think some ablation studies will be good to add. For example, what if only using StreamingLLM as the draft model, or only using the JF68M model as the draft model.
- The approach, while being very efficient for 128K-token sequences. However, it may not further extend to much longer input (e.g., 1M), as the memory will be the bottleneck in that case.

---

> ### Author Rebuttal · Authors · 2024-05-31
>
> Thank you for your encouraging feedback and insightful remarks. We have carefully addressed your questions and hope you will consider raising your score based on our response.
>
> **Q1: Clarification of Figure 3b**
>
> “Top-K” means taking the subset of KV cache with top-K largest attention weights. We will define “acceptance rate” clearly in the revised paper.
>
> **Q2: Clarification of “maintain a full cache for the initial two layers”**
>
> As shown in Figure 3a, the first two layers are less sparse. Hence, we did not sparsify them in the observation section. For system efficiency, we sparsify all layers in the implemented system. This may lower the acceptance rate, but the constant drafting cost makes it a worthwhile trade-off.
>
> **Q3: Clarification of Figure 3c**
>
> The x-axis represents the index of generated tokens. “Recovery rate” refers to the ratio of retrieved attention scores to the ground truth sum of attention scores for the prompt tokens ($s_1, \ldots, s_n$). Assume the generated token index is $m$:
>
> $$\text{Recovery rate} = \frac{\sum_{s_{mi} \in R} s_{mi}}{\sum_{i=1}^n s_{mi}}$$
>
> where $R$ represents the retrieved attention scores among prompt tokens.
>
> **Q4: Clarification of “JF68M Eviction”**
>
> “JF68M Eviction” refers to using only the JF68M model with StreamingLLM as the draft model.
>
> **Q5: Small draft model concern**
>
> TriForce is a general method for long sequence generation. Many existing self-speculative models can replace the small model for hierarchical speculation, such as Medusa [1] or EAGLE [2].
>
> **Q6: Ablation studies**
>
> We clarify that our paper includes some ablation studies in Table 2. For example, “StreamingLLM w/ Hierarchy” represents replacing retrieval-based drafting with StreamingLLM. Additionally, we provide a table here.
> |Method|Speedup|
> |-|-|
> |Retrieval w/o Hierarchy|1.80x|
> |StreamingLLM w/o Hierarchy|1.75x|
> |StreamingLLM w/ Hierarchy|1.90x|
> |JF68M StreamingLLM|1.42x|
>
> **Q7: Scalability**
>
> For longer inputs, a solution is offloading the KV cache to CPU memory. Below are experiments on L40 with offloading for longer inputs on LWM [3] models using PG-19, showing impressive speedup.
> |Model|Speedup|
> |-|-|
> |LWM-Text-256K|11.81x|
> |LWM-Text-512K|12.10x|
>
> [1] Medusa: Simple LLM Inference Acceleration Framework with Multiple Decoding Heads
>
> [2] EAGLE: Speculative Sampling Requires Rethinking Feature Uncertainty
>
> [3] World Model on Million-Length Video And Language With Blockwise RingAttention

---

> > ### Comment · Reviewer_myEE · 2024-06-06
> >
> > Thank authors for the response. I will keep my score as it is.

---

### Official Review · Reviewer_ByTv · 2024-05-16

**Rating:** 7
**Confidence:** 3
**Ethics Flag:** 1

**Summary:**

The paper introduces TRIFORCE, a novel speculative decoding system aimed at accelerating long-sequence generation in large language models (LLMs). The proposed method integrates Retrieval-based Drafting and Hierarchical Speculation to tackle the dual bottlenecks of model weights and key-value (KV) cache. TRIFORCE leverages the original model weights and partial KV cache to create a draft model, which is further speculated by a lightweight model using StreamingLLM cache. The approach is shown to provide significant speedups, particularly for the Llama2-7B-128K model, without compromising the quality of the generated sequences.

**Reasons To Accept:**

1. The proposed method is straightforward and easy to follow.
2. Using partial KV cache as a draft model is novel in the field of speculative decoding.
3. The method indeed shows notable speedup, particularly in long context scenarios.

**Reasons To Reject:**

The main weakness is the lack of comparisons or discussions of related work:
1. The Hierarchical Speculation in Section 4.2 is not novel. [1] has proposed similar ideas but is not compared or discussed in related work (Section 2.1).
2. [2] shows similar ideas as Retrieval-based Drafting in section 4.1, but is not compared or discussed in related work.
3. Skipping layers[3] as a draft model is a similar method to your using partial KV cache, but is not compared or discussed in related work.

My intention is not that TriForce should beat these methods, since using partial KV cache as a draft model is new and should be useful in some scenarios (e.g. long context). However, I wonder why these related works are missed. I would raise my score if the authors justify the reasons.

[1] Accelerating LLM Inference with Staged Speculative Decoding

[2] REST: Retrieval-Based Speculative Decoding

[3] Draft & Verify: Lossless Large Language Model Acceleration via Self-Speculative Decoding

---

> ### Author Rebuttal · Authors · 2024-05-31
>
> Thank you for detailed review and valuable feedback. We appreciate the reviewer’s recognition of the novelty and effectiveness of our method. Below, we address concerns regarding the lack of comparisons or discussions of related work. We hope the reviewer can consider raising your score in light of our response.
>
> **Q1: Discussion of Staged Speculative Decoding**
>
> Thank you for the helpful suggestion. We will add a detailed discussion to the revised paper as below.
>
> “Staged speculation techniques [1] have been suggested to further accelerate inference by using a cascade of draft models, and hierarchical speculation shares similarities with these approaches. However, our method focuses on long sequence generation tasks, which presents unique challenges. Our approach uses different drafting methods for each speculation phase to address two distinct bottlenecks in long-context scenarios: model weights and KV cache. This differs from [1], which focuses on solving the same bottleneck (model weights) at every stage for further acceleration.”
>
> **Q2&Q3: Comparison and Discussion of Retrieval-based Drafting and Skipping Layers**
>
> We appreciate the suggestion to compare TriForce with these two methods. The table below compares TriForce, REST [2], and Skipping Layers [3] with Llama2-7B-128K on an A100 GPU using the PG-19 dataset, showing that TriForce achieves the best speedup for long sequence generation settings.
> |Method|Speedup|
> |-|-|
> |TriForce|2.31x|
> |REST [2]|1.47x|
> |Skipping Layers [3]|1.36x|
>
> In addition to the efficiency comparison, we also discuss each method as follows:
>
> **REST [2]**: Our retrieval-based drafting differs from REST, as our method retrieves information from the KV cache, while REST retrieves tokens from a predefined datastore. This distinction allows our approach to dynamically adapt to the context during generation, rather than relying on an external datastore.
>
> **Skipping Layers [3]**: In long-context settings where the primary bottleneck is the KV cache, this method shows some limitations. In the table above, it utilizes 68% of the KV cache, which was found to be the optimal combination of skipped layers. In contrast, TriForce can efficiently use just 3% of the KV cache, better addressing the bottleneck in long-context scenarios.
>
> [1] Accelerating LLM Inference with Staged Speculative Decoding
>
> [2] REST: Retrieval-Based Speculative Decoding
>
> [3] Draft & Verify: Lossless Large Language Model Acceleration via Self-Speculative Decoding

---

### Decision · Program_Chairs · 2024-07-10

**Decision:**

Accept

**Comment:**

### Summary
The paper "TriForce: Lossless Acceleration of Long Sequence Generation with Hierarchical Speculative Decoding" introduces a novel approach aimed at improving the efficiency of long-sequence generation in large language models (LLMs). The method, named TriForce, combines Retrieval-based Drafting and Hierarchical Speculation to address the bottlenecks associated with model weights and key-value (KV) cache. The authors demonstrate significant speedups, particularly for the Llama2-7B-128K model, without compromising the quality of the generated sequences.

### Strengths
1. **Innovative Use of KV Cache**: The use of partial KV cache as a draft model is a novel idea in the field of speculative decoding.
2. **Significant Speedup**: The method shows notable speedup, especially in long-context scenarios, achieving up to 2.31x speedup on A100 GPUs and 11.65x in offloading settings on an L40 GPU.
3. **Strong Experimental Results**: The experimental results are robust and demonstrate the effectiveness of the method across various temperatures and settings.

### Weaknesses
1. **Lack of Comparisons to Related Work**: Initial submission lacked comparisons or discussions of related work. This was addressed in the rebuttal, but it would have been beneficial to include these comparisons in the original submission.
2. **Presentation and Clarity**: Some parts of the paper were hard to follow without familiarity with related work and concepts. The authors have addressed this in their rebuttal, clarifying key terms and concepts.

### Recommendations for Authors
1. **Include Comprehensive Related Work**: Ensure that the final manuscript includes detailed comparisons and discussions of related work as provided in the rebuttal.
2. **Clarify Key Concepts**: Improve the clarity of the presentation by defining key terms and concepts, as suggested by the reviewers.

### Final Decision
The paper presents a significant contribution to the field of long-sequence generation in LLMs with its use of hierarchical speculative decoding and retrieval-based drafting. The method's effectiveness is well-supported by strong experimental results. Despite some initial weaknesses in the presentation and lack of comparisons, the authors have addressed these concerns adequately in their rebuttal. Therefore, I recommend accepting this paper for COLM 2024.